# Variation in Protein and Isoflavone Contents of Collected Domestic and Foreign Soybean (*Glycine max* (L.) Merrill) Germplasms in Korea

Ji-Seok Lee, Hong-Sig Kim  and Tae-Young Hwang *

Department of Crop Science, College of Agriculture, Life Science and Environmental Chemistry, Chungbuk National University, Cheongju 28644, Korea; haru412@daum.net (J.-S.L.); hongsigk@chungbuk.ac.kr (H.-S.K.)
* Correspondence: hwangty@chungbuk.ac.kr

**Abstract:** This study was carried out to investigate the variations in protein and isoflavone contents of 300 soybean germplasms introduced from domestic and foreign countries and to compare their contents in terms of size, colour and country of origin. The protein content ranged from 28.7 g 100 g$^{-1}$ to 44.5 g 100 g$^{-1}$, with an average of 39.0 g 100 g$^{-1}$. In a comparison of protein according to country of origin, the highest content was seen in soybeans from Korea (39.7 g 100 g$^{-1}$), followed by North Korea (39.2 g 100 g$^{-1}$), China (39.0 g 100 g$^{-1}$), Japan (38.8 g 100 g$^{-1}$), the USA (38.0 g 100 g$^{-1}$) and Russia (37.2 g 100 g$^{-1}$). The total isoflavone content ranged from 207.0 μg g$^{-1}$ to 3561.8 μg g$^{-1}$, with an average of 888.8 μg g$^{-1}$. In the comparison of isoflavone content according to country, the highest average content was shown in soybeans from Japan (951.3 μg g$^{-1}$), followed by the USA (918.7 μg g$^{-1}$), Korea (902.2 μg g$^{-1}$), North Korea (870.0 μg g$^{-1}$) and Russia (710.6 μg g$^{-1}$). Daidzein, glycitein and genistein isoflavone contents were positively correlated, while total isoflavone and protein showed a low negative correlation.

**Keywords:** soybean germplasm; protein content; isoflavone content; daidzein; glycitein; genistein

## 1. Introduction

Soybean germplasm pools in Asia are divided into seven groups defined by geographic region: northeastern China and Siberia, central and southern China, Korea, Japan, Taiwan and southern Asia, northern India and Nepal, and central India [1]. The soybean plant originated in northeastern China and the Korean peninsula around BC 1700–1100 and became widely distributed throughout East Asia, including Japan and the Maritime Province of Siberia. It is estimated that it then gradually spread to southeast Asian regions, such as Vietnam, Thailand, Malaysia and Myanmar [2]. Soybean has become an important food for a diversity of cultures, and over long periods various kinds of domestic cultivars have been differentiated and cultivated to adapt the plant to different environments, including regions of Korea [3–5]. Korean domestic soybean cultivars reportedly have a high level of genetic variation [6–10], and promising domestic soybean cultivars have been used as parent lines and have contributed greatly to improving crop quality and yield [11]. It is clear that soybean genetic resources are important in breeding programmes [6,12].

Soybean seeds contain about 40 g 100 g$^{-1}$ protein [13,14], within a range of 25 g·100 g$^{-1}$ to 45 g 100 g$^{-1}$, and the main fractions consist of 11S (glycinin) and 7S (*β*-conglycinin) [15]. Soybean proteins have excellent nutritional and physicochemical functions compared with other plant proteins [16]. Therefore, the goal of soybean breeding programmes is to develop new varieties with increased protein content and quality and high amounts of functional components [14]. Soybean cultivars show greater variation in protein content than the variation caused by the environment [17–19] and usually accumulate more protein during the reproductive stages when grown at low temperatures [18–21]. Soybean genetic resources show protein content variation in the region of 34 g 100 g$^{-1}$ to 57 g 100 g$^{-1}$

compared to dry weight [21,22], and the high protein content of soybeans increases the yield when they are processed into pastes such as tofu and soybean paste [23,24].

Soybean seeds contain approximately 0.2% to 0.4% isoflavones [25], and the content varies greatly depending on the variety and the cultivation environment. Even within the same variety, there are substantial variations in the content of isoflavones depending on the year and region of cultivation [25,26]. The isoflavone content decreases in regions with high rainfall [27,28]; the average isoflavone content of seeds from four domestic varieties of soybean cultivated in paddy fields was 992 mg/kg, which is 237 mg/kg higher than those grown in dry fields [29]. The isoflavones produced by soybean include three substances and four chemical structures. The 12 isomers exist in forms such as aglycones (daidzein, glycitein, genistein), glucosides (daidzin, glycitin, genistin), malonylglucoside derivatives (6"-*O*-malonyldaidzin, 6"-*O*-malonylglycitin, 6"-*O*-malonylgenistin) and acetylglucoside derivatives (6"-*O*-acetyldaidzin, 6"-*O*-acetylglycitin, 6"-*O*-acetylgenistin) [28,30]. Soybean isoflavones are phytoestrogens structurally similar to 17-β-oestradiol that have the same effect on oestrogen receptors (ER-α and ER-β) as the female hormone oestrogen, reducing blood cholesterol and inhibiting skeletal loss to prevent cardiovascular disease and osteoporosis in menopausal women and promoting high physiological activity to relieve postmenopausal syndrome and prevent breast, prostate, ovarian and colon cancers [31]. In addition, cosmetics containing soybean isoflavones reduce the need for dihydrotestosterone, which can lead to acne vulgaris lesions [32]. Isoflavones are absorbed by the body to different degrees according to their form. Glucosides are barely absorbed by the intestine, whereas, when aglycones are hydrolysed by β-glucosidase, they are rapidly absorbed by passive diffusion [33,34]. Two absorbable isoflavones, genistein and daidzein, are known as representative isoflavones with excellent physiological functions [35–38]. The isoflavones in non-fermented soybean-based foods are mostly biologically inactive β-glucosides, whereas in fermented soybean foods, they are typically in the form of aglycones, e.g., genistein, daidzein and glycitein, which account for about 50%, 40% and 10%, respectively.

Since the 2000s, consumer demand has diversified, and there has been a rapid increase in interest in health and food functionality; therefore, the breeding goals of soybean producers are functionality, safety, high quality and diversity of use. Soybean seeds have excellent nutritional benefits such as high-quality protein, essential fatty acids and other physiologically active substances, including isoflavones, saponins, phytic acid, vegetable sterol, dietary fibre and protease inhibitors. Interest in isoflavones as important anticancer substances is also emerging. Many research findings on soybean protein and isoflavones have been reported. However, research to discover useful genetic resources for the development of high-quality, high-protein and high-isoflavone varieties is still insufficient. Therefore, this study was conducted to investigate the variation in protein and isoflavone contents of soybean germplasms and to select useful genetic resources for the development of new varieties.

## 2. Materials and Methods

### 2.1. Plant Materials and Cultivation

We examined 300 soybean germplasms collected from six countries (Korea, 117 genetic germplasms; China, 71; Japan, 46; USA, 43; Russia, 12; and North Korea, 11) (Table 1). Seed size was classified into large (>24 g), medium (13–24 g) and small (<13 g) by 100-seed weight. Seed colour was divided into black, green, yellow and brown. Three hundred soybean germplasms were cultivated in an experimental field of Chungbuk National University after sowing on 27 May 2019. The density was 70 cm (row) × 15 cm (plant) with three seeds; two seedlings were thinned as the first leaf emerged. Fertilizer was supplied at 50 kg/km$^2$ (N:P$_2$O$_5$:K$_2$O = 3:3:3.4 kg/ha), and other cultivation management strategies were in accordance with standard soybean cultivation methods based on Rural Development Administration recommendations. The harvested soybean seeds were stored in a −20 °C freezer and used for protein and isoflavone analysis.

**Table 1.** Number of 300-soybean germplasms classified by collected country, seed size and colour.

| | | |
|---|---|---|
| Collected country | KOR [†] | n = 117 |
| | CHN | n = 71 |
| | JPN | n = 46 |
| | USA | n = 43 |
| | RUS | n = 12 |
| | NK | n = 11 |
| | Total | n = 300 |
| Seed size | Large [‡] | n = 107 |
| | Medium | n = 153 |
| | Small | n = 40 |
| | Total | n = 300 |
| Seed colour | Black | n = 37 |
| | Brown | n = 21 |
| | Green | n = 50 |
| | Yellow | n = 192 |
| | Total | n = 300 |

[†] KOR: Korea, CHN: China, JPN: Japan, USA: United States of America, RUS: Russia, NK: North Korea. [‡] Large (100 seed weight: >24 g), Medium (100 seed weight: 13 to 24 g), Small (100 seed weight: <13 g).

### 2.2. Seed Protein Extraction

The soybean seeds were harvested and dried for 3 days in a 40 °C dry oven and then analysed after confirming that the moisture content was 16% or less. The protein content of the seeds was analysed by the micro-Kjeldahl method [39]. Soybean seeds were pulverised using a grinder (UDY Co., Fort Collins, CO, USA), and 50 mg of the powder was hydrolysed with 20 mL of concentrated sulphuric acid ($H_2SO_4$) and catalyst ($CuSO_4$ + $K_2SO_4$) in a digester (Hanil Lab Tech Co., Yangju, Korea) at 415 °C for 2 h. After cooling at room temperature for 2 h, the product was distilled using an automatic Kjeldahl distillation unit (Hanil Lab Tech Co., Yangju, Korea) and titrated with a standard solution of ammonia sulphuric acid (0.1 N) adsorbed on boric acid. The amount of total nitrogen in the raw material was multiplied by the traditional conversion factor of 6.25 to determine the total protein content [39].

### 2.3. Isoflavones Extraction and HPLC Analysis

For the isoflavone analysis, daidzein, glycitein and genistein standards were purchased from Sigma-Aldrich (St. Louis, MO, USA), dissolved in 100% dimethylsulphoxide (DMSO) and diluted to five different concentrations according to the concentration gradient. In the chromatogram obtained by injecting 20 µL into a high-performance liquid chromatography (HPLC) column, the calibration curve equation and coefficient of determination between the measured value of the peak area and the concentration of the standard solution were calculated (Table S1).

The isoflavones from soybean seeds (50 mg of powder) were first extracted using 15 mL of 1 N HCl and acid-hydrolysed in an oven at 105 °C for 2 h to convert the glycosides into the aglycone forms. The mixtures were cooled at room temperature for 1 h, 20 mL of 100% MeOH was added and the aglycones were extracted by stirring for 2 h. We added 100% MeOH to the extract and passed it through a 0.45 µm filter before HPLC analysis. The HPLC analysis was performed with a Waters 600 series pump and controller, and Waters 486 tunable absorbance detector (Waters Co., Milford, MA, USA). Twenty microliters of the sample was injected into a YMC-PACK Pro C18 column (250 × 4.6 mm i.d.) (YMC Co., Kyoto, Japan) using a 30 min. linear gradient of 20–50% acetonitrile (*v*/*v*) in aqueous solution containing constant 0.1% acetic acid. The UV absorption was measured at 254 nm (Table S2). The aglycones were separated in the order of daidzein, glycitein and genistein according to retention time (Figure S1).

### 2.4. Data Analysis

All experiments were conducted in triplicate, and the data are reported as mean ± standard deviation. All data were subjected to analysis of Duncan's multiple range test (DMRT) via the SAS software package (release 9.4; SAS Institute, Cary, NC, USA). Data were analysed using the PROC general linear model (GLM) procedure, and means were separated on the basis DMRT. Significances were set at the 5% level. Soybean germplasms classified based on the contents of protein and isoflavone were analysed according to collected country, seed size and seed colour using multivariate analysis.

## 3. Results

### 3.1. Protein Content of Soybean Seeds

The protein content of the seeds was 28.7 g 100 g$^{-1}$ to 44.57 g 100 g$^{-1}$ (average 39.07 g 100 g$^{-1}$) (Table S3). Of the 300 soybean germplasms, the majority (99.3 g 100 g$^{-1}$) ranged between 38 g 100 g$^{-1}$ and 40 g 100 g$^{-1}$ protein (Figure 1), and higher protein contents ($\geq$ 43.5 g 100 g$^{-1}$) were seen in IT263050 (44.5 g 100 g$^{-1}$), IT101111 (44.0 g 100 g$^{-1}$), IT24921 (43.8 g 100 g$^{-1}$), IT167907 (43.8 g 100 g$^{-1}$), IT102595 (43.6 g 100 g$^{-1}$) and IT161574 (43.5 g 100 g$^{-1}$) (Table S4). IT263050, IT24921 and IT161574 were genetic resources collected in China, and IT101111, IT167907 and IT102595 were obtained from Korea.

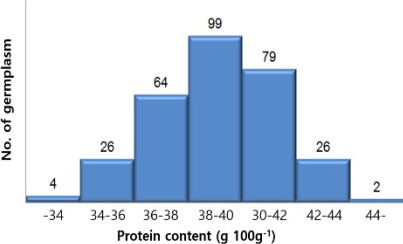

**Figure 1.** Frequency distributions of protein contents in 300 soybean germplasms.

Soybean seeds from Korea had the highest protein content at 39.7 g 100 g$^{-1}$, followed by North Korea (39.2 g 100 g$^{-1}$), China (39.0 g 100 g$^{-1}$), Japan (38.8 g 100 g$^{-1}$), the United States (38.0 g 100 g$^{-1}$) and Russia (37.2 g 100 g$^{-1}$) (Table 2). The difference in protein content among the countries was highly significant, with a $p$-value < 0.001. The germplasm with the highest protein content was IT263050 from China, containing 44.5 g 100 g$^{-1}$, and the germplasm with the lowest content was IT156248 (28.7 g 100 g$^{-1}$), which was collected in Japan.

**Table 2.** Maximum, minimum and mean value of protein contents according to the collected country in 300 soybean germplasms.

|  | Country | No. | Max. | Min. | Mean | ±SD | CV (%) |
|---|---|---|---|---|---|---|---|
|  | CHN [†] | n = 71 | 44.5 | 33.6 | 39.0 [b,‡] | 2.2 | 5.7 |
|  | JPN | n = 46 | 42.3 | 28.7 | 38.8 [b] | 2.5 | 6.6 |
| Protein | KOR | n = 117 | 44 | 35.5 | 39.7 [a] | 2.3 | 5.7 |
| (g 100 g$^{-1}$) | NK | n = 11 | 42.6 | 34.9 | 39.2 [b] | 1.9 | 5.0 |
|  | RUS | n = 12 | 41 | 34.5 | 37.2 [d] | 2.2 | 5.9 |
|  | USA | n = 43 | 42.8 | 31.5 | 38.0 [c] | 2.4 | 6.4 |
| $p$-value |  |  |  | <0.001 |  |  |  |

[†] KOR: Korea, CHN: China, JPN: Japan, USA: United States of America, RUS: Russia, NK: North Korea. [‡] Means followed by the same letter in each column are not significantly different by Duncan's multiple range test at 5% level.

In Korea, soybean seeds are generally classified into three sizes, small (<12 g), medium (12–24 g) and large (>24 g), based on their 100-seed weight. In terms of changes in protein content with seed size, the highest average of 39.7 g 100 g$^{-1}$ was observed in small seed germplasms, followed by large seed germplasms at 39.3 g 100 g$^{-1}$ and medium-sized

germplasms at 38.6 g 100 g$^{-1}$ (Table 3). The difference in protein content according to seed size was recognized as significant with a *p*-value of 0.008, and there was also a significant difference according to the results of DART.

**Table 3.** Maximum, minimum and mean value of protein contents according to seed sizes in 300 soybean germplasms.

|  | Size | No. | Max. | Min. | Mean | ±SD | CV (%) |
|---|---|---|---|---|---|---|---|
| Protein (g 100 g$^{-1}$) | Large ‡ | n = 107 | 44.0 | 35.3 | 39.3 b,† | 2.0 | 5.2 |
|  | Medium | n = 153 | 44.5 | 28.7 | 38.6 c | 2.5 | 6.5 |
|  | Small | n = 40 | 43.8 | 34.3 | 39.7 a | 2.6 | 6.6 |
| *p*-value |  |  |  |  | 0.008 |  |  |

† Means followed by the same letter in each column are not significantly different by Duncan's multiple range test at 5% level. ‡ Large (100 seed weight: >24 g), Medium (100 seed weight: 13 to 24 g) and Small (100 seed weight: <13 g).

Differences in protein content according to seed colour were found in the order of yellow-seed germplasms (39.1%), green seed germplasms (39.0%), black seed germplasms (38.9%) and brown seed germplasms (38.6%), but the differences were not significant (Table 4).

**Table 4.** Maximum, minimum and mean value of protein contents according to seed colour in 300 soybean germplasms.

|  | Colour | No. | Max. | Min. | Mean | ±SD | CV (%) |
|---|---|---|---|---|---|---|---|
| Protein (g 100 g$^{-1}$) | Black | n = 37 | 44.5 | 34.3 | 38.9 a,† | 2.3 | 5.8 |
|  | Brown | n = 21 | 42.1 | 34.4 | 38.6 a | 2.5 | 6.4 |
|  | Green | n = 50 | 42.9 | 34.7 | 39.0 a | 2.2 | 5.5 |
|  | Yellow | n = 192 | 44.0 | 28.7 | 39.1 a | 2.5 | 6.4 |
| *p*-value |  |  |  |  | 0.860 |  |  |

† Means followed by the same letter in each column are not significantly different by Duncan's multiple range test at 5% level.

### 3.2. Isoflavone Contents of Soybean Seeds

The isoflavone contents of the 300 soybean germplasms collected from six counties are shown in Table S5. The seeds had 207.0–3561.8 μg g$^{-1}$ of isoflavone (average = 888.8 μg g$^{-1}$ seed). In terms of each component, the average content of daidzein was 338.4 μg g$^{-1}$ and ranged from 56.4 to 2081.4 μg g$^{-1}$, and that of glycitein was 126.0 μg g$^{-1}$ and ranged from 17.7 to 443.7 μg g$^{-1}$. The average genistein content was 424.2 μg g$^{-1}$ and ranged from 28.2 to 1378.4 μg g$^{-1}$. Genistein was the most abundant isoflavone, followed by daidzein and glycitein. In most soybean germplasms, the content of daidzein and genistein was about 90% of the total isoflavone content. The content distributions of isoflavones in the 300 collected soybean germplasms are shown in Figure 2. Daidzein was found in 137 germplasms at the range of 200–400 μg g$^{-1}$, 75 germplasms at <200 μg g$^{-1}$ and 5 germplasms at >1000 μg g$^{-1}$. Glycitein concentrations of <100 μg g$^{-1}$ were observed in 112 germplasms, and >300 μg g$^{-1}$ was seen in 11 germplasms. Most of the germplasms (121) had in the range of 200–400 μg g$^{-1}$, 28 germplasms had <200 μg g$^{-1}$ and 2 germplasms had >1000 μg g$^{-1}$. The distribution of the total isoflavone content was 165 germplasms in the range of 500–1000 μg g$^{-1}$ and 77 germplasms in the range of 1000–1500 μg g$^{-1}$, and more than 80% of germplasms had between 500 μg g$^{-1}$ and 1500 μg g$^{-1}$. Six germplasms, IT262889 (3561.8 μg g$^{-1}$), IT171009 (2271.0 μg g$^{-1}$), IT100869 (2250.5 μg g$^{-1}$), IT208248 (2179.3 μg g$^{-1}$), IT142911 (2028.7 μg g$^{-1}$) and IT142854 (2017.1 μg g$^{-1}$), showed a total isoflavone content of >2000 μg g$^{-1}$ (Table S6), of which IT262889 collected in Japan showed the highest total isoflavone content.

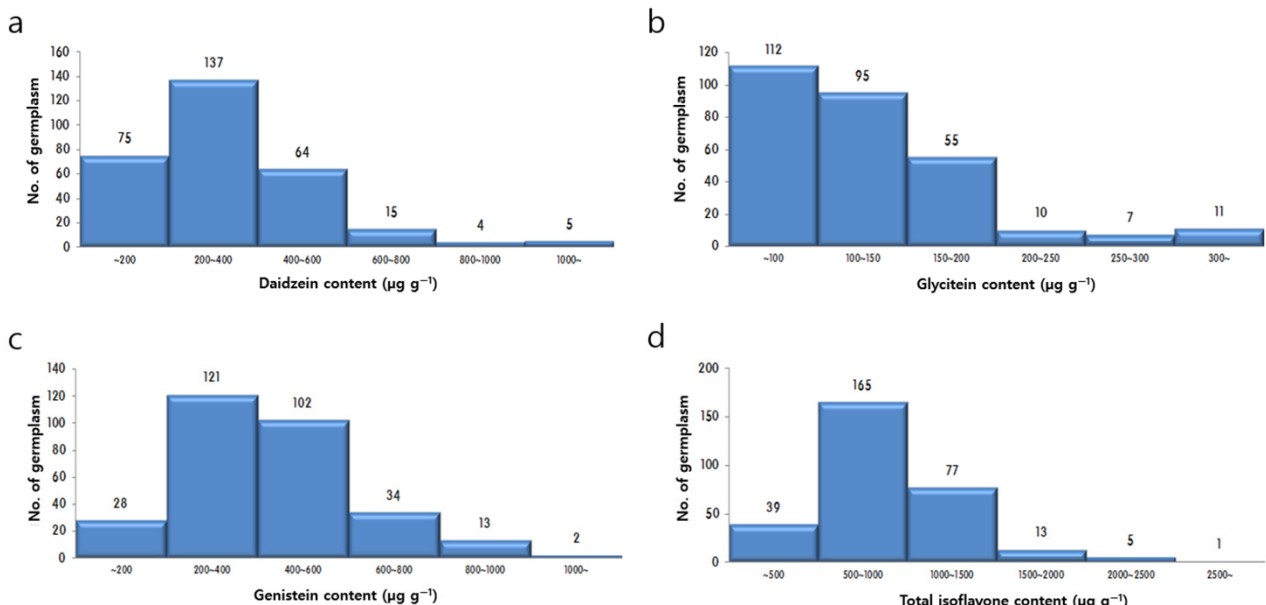

**Figure 2.** Frequency distributions of daidzein (**a**), glycitein (**b**), genistein (**c**) and total isoflavone (**d**) contents in 300 soybean germplasms.

The average total isoflavone content by country of collection was 951.3 μg g$^{-1}$ in Japan, 918.7 μg g$^{-1}$ in the United States, 902.2 μg g$^{-1}$ in Korea, 870.0 μg g$^{-1}$ in North Korea, 841.3 μg g$^{-1}$ in China and 710.6 μg g$^{-1}$ in Russia, but there were no significant differences. The daidzein and genistein contents also did not differ significantly between countries, but the glycitein content showed a high level of significance, with a *p*-value of less than 0.001 (Table 5).

When we compared the contents of various seed sizes, the total isoflavone content was the highest in small seeds with an average of 1156.0 μg g$^{-1}$, followed by large seeds with an average of 898.4 μg g$^{-1}$ and medium-sized seeds with an average of 812.3 μg g$^{-1}$. The isoflavone content was significantly different between seed sizes, with a *p*-value of <0.001, and there was also a significant difference in the DART results (Table 6).

Differences in total isoflavone content according to seed colour were significant with a *p*-value of <0.001 (Table 7). The average isoflavone content was 1074.2 μg g$^{-1}$ in brown germplasms, 1070.1 μg g$^{-1}$ in black seed germplasms, 1006.2 μg g$^{-1}$ in green germplasms and 803.0 μg g$^{-1}$ in yellow germplasms. The daiszein content was the highest in black germplasms (436.5 μg g$^{-1}$) compared with green germplasms (406.4 μg g$^{-1}$), brown germplasms (402.7 μg g$^{-1}$) and yellow germplasms (294.9 μg g$^{-1}$). The glycitein content was the highest in germplasms (158.4 μg g$^{-1}$) compared with black germplasms (152.4 μg g$^{-1}$), yellow germplasms (121.5 μg g$^{-1}$) and green germplasms (110.1 μg g$^{-1}$). The genistein content was the highest in brown germplasms (513.1 μg g$^{-1}$) compared with green germplasms (489.7 μg g$^{-1}$), black germplasms (481.3 μg g$^{-1}$) and yellow germplasms (386.6 μg g$^{-1}$).

The genetic distances between all pairings of groups by release period are shown in Table 7. The genetic distance between varieties developed after 2000 and in the 1980s was the furthest at 0.5731, and the genetic distance between varieties developed after 2000 and in the 1990s was the nearest at 0.1909 (Table 7).

### 3.3. Correlation between Protein and Isoflavone Contents of Soybean Seeds

The correlations between protein and dadzein, glycitein, genistein and total isoflavone contents are shown in Figure 3. The correlation coefficients for protein compared to daizein, glycitein or genistein contents were −0.1694, −0.1706 and −0.222, respectively, and between the protein and total isoflavone content, the coefficient was −0.2200. Therefore, there was a slight negative correlation between the protein content and that of isoflavones.

**Table 5.** Maximum, minimum and mean value of isoflavone contents according to collected countries in 300 soybean germplasms.

| Country | Statistics | Isoflavone Contents ($\mu g\ g^{-1}$) | | | |
| --- | --- | --- | --- | --- | --- |
| | | Daidzein | Glycitein | Genistein | Total |
| KOR [†] (n = 117) | Max. | 958.7 | 380.2 | 882.1 | 1973.7 |
| | Min. | 56.5 | 17.7 | 121.7 | 267.7 |
| | Mean | 352.2 [a,‡] | 109.7 [d] | 440.3 [a] | 902.2 [a] |
| CHN (n = 71) | Max. | 1016.0 | 351.6 | 949.1 | 2271.1 |
| | Min. | 84.0 | 45.2 | 95.1 | 278.2 |
| | Mean | 302.5 [a] | 151.6 [a] | 387.2 [a] | 841.3 [a] |
| JPN (n = 46) | Max. | 2081.5 | 302.0 | 1378.5 | 3561.9 |
| | Min. | 75.0 | 34.5 | 28.3 | 207.0 |
| | Mean | 378.9 [a] | 103.6 [e] | 468.8 [a] | 951.3 [a] |
| USA (n = 43) | Max. | 1060.4 | 443.8 | 906.2 | 2179.4 |
| | Min. | 62.3 | 56.8 | 141.0 | 260.1 |
| | Mean | 354.6 [a] | 152.5 [a] | 411.6 [a] | 918.7 [a] |
| RUS (n = 12) | Max. | 331.0 | 174.6 | 479.2 | 911.8 |
| | Min. | 150.3 | 72.9 | 191.1 | 491.0 |
| | Mean | 251.8 [a] | 131.7 [b] | 327.1 [a] | 710.6 [a] |
| NK (n = 11) | Max. | 498.0 | 245.0 | 923.1 | 1415.7 |
| | Min. | 124.6 | 40.4 | 214.9 | 412.9 |
| | Mean | 288.2 [a] | 119.4 [c] | 462.5 [a] | 870.0 [a] |
| *p*-value | | 0.1893 | <0.001 | 0.1030 | 0.4619 |

[†] KOR: Korea, CHN: China, JPN: Japan, USA: United States of America, RUS: Russia, NK: North Korea. [‡] Means followed by the same letter in each column are not significantly different by Duncan's multiple range test at 5% level.

**Table 6.** Maximum, minimum and mean value of isoflavone contents according to seed sizes in 300 soybean germplasms.

| Seed Size (100-Seed Weight) | Statistics | Isoflavone Contents ($\mu g\ g^{-1}$) | | | |
| --- | --- | --- | --- | --- | --- |
| | | Daidzein | Glycitein | Genistein | Total |
| Small [‡] n = 40 | Max. | 1060.4 | 443.8 | 949.1 | 2271.1 |
| | Min. | 56.5 | 57.5 | 153.4 | 267.7 |
| | Mean | 477.6 [a,†] | 197.9 [a] | 480.5 [a] | 1156.0 [a] |
| | ±SD | 259.3 | 104.5 | 204.4 | 522.2 |
| | CV (%) | 54.3 | 52.8 | 42.5 | 45.2 |
| Medium n = 153 | Max. | 2081.5 | 265.7 | 1378.5 | 3561.9 |
| | Min. | 62.3 | 34.6 | 65.0 | 207.0 |
| | Mean | 295.1 [c] | 122.1 [b] | 395.0 [c] | 812.3 [c] |
| | ±SD | 200.9 | 53.5 | 199.0 | 390.0 |
| | CV (%) | 68.1 | 43.9 | 50.4 | 48.0 |
| Large n = 107 | Max. | 1018.8 | 271.5 | 1144.7 | 2250.5 |
| | Min. | 97.9 | 17.7 | 28.3 | 240.5 |
| | Mean | 348.5 [b] | 104.8 [c] | 445.1 [b] | 898.4 [b] |
| | ±SD | 166.5 | 46.0 | 186.6 | 348.3 |
| | CV (%) | 47.8 | 43.9 | 41.9 | 38.8 |
| *p*-value | | <0.001 | <0.001 | 0.0198 | <0.001 |

[†] Means followed by the same letter in each column are not significantly different by Duncan's multiple range test at 5% level. [‡] Large (100 seed weight: >24 g), Medium (100 seed weight: 13 to 24 g), Small (100 seed weight: <13 g).

**Table 7.** Maximum, minimum and mean value of isoflavone contents according to seed colour in 300 soybean germplasms.

| Seed Colour | Statistics | Isoflavone Contents ($\mu$g g$^{-1}$) | | | |
|---|---|---|---|---|---|
| | | Daidzein | Glycitein | Genistein | Total |
| Black n = 37 | Max. | 1060.4 | 380.2 | 859.5 | 2028.8 |
| | Min. | 56.5 | 36.2 | 106.4 | 240.5 |
| | Mean | 436.5 [a,†] | 152.4 [b] | 481.3 [c] | 1070.1 [a] |
| | ±SD | 244.9 | 95.0 | 174.7 | 459.5 |
| | CV (%) | 56.1 | 62.3 | 36.3 | 42.9 |
| Brown n = 21 | Max. | 1016.0 | 443.8 | 949.1 | 2271.1 |
| | Min. | 62.3 | 17.7 | 141.0 | 260.1 |
| | Mean | 402.7 [b] | 158.4 [a] | 513.1 [a] | 1074.2 [a] |
| | ±SD | 218.5 | 218.5 | 214.9 | 506.8 |
| | CV (%) | 54.3 | 54.3 | 41.9 | 47.2 |
| Green n = 50 | Max. | 688.0 | 213.7 | 918.6 | 1752.1 |
| | Min. | 108.3 | 40.0 | 99.6 | 251.1 |
| | Mean | 406.4 [b] | 110.1 [d] | 489.7 [b] | 1006.2 [b] |
| | ±SD | 126.5 | 40.5 | 144.7 | 256.6 |
| | CV (%) | 31.1 | 36.8 | 29.6 | 25.5 |
| Yellow n = 192 | Max. | 2081.5 | 678.2 | 1378.5 | 3561.9 |
| | Min. | 75.0 | 34.5 | 28.3 | 207.0 |
| | Mean | 294.9 [c] | 121.5 [c] | 386.6 [d] | 803.0 [c] |
| | ±SD | 202.4 | 54.8 | 202.2 | 398.3 |
| | CV (%) | 68.6 | 45.1 | 52.3 | 49.6 |
| | *p*-value | <0.001 | 0.0022 | <0.001 | <0.001 |

[†] Means followed by the same letter in each column are not significantly different by Duncan's multiple range test at the 5% level.

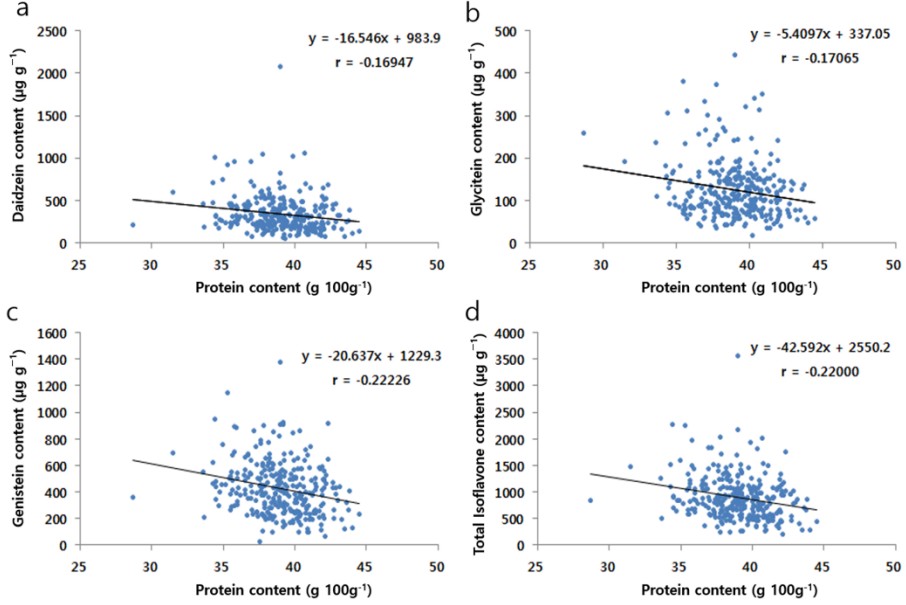

**Figure 3.** Relationship among daidzein (**a**), glycitein (**b**), genistein (**c**) and total isoflavone (**d**) and protein contents in 300 soybean germplasms.

## 4. Discussion

Wilson [21] reported that the protein content of soybean seeds was distributed in the range of 34% to 57% of dry matter weight. In this study, the protein content of the collected soybean germplasms was in the range of 29% to 45%. The protein content was in the range of 38–40% in 99 soybean germplasms, 34% or less in four soybean germplasms, and

44% or more in two soybean germplasms (Figure 1). Kumar [40] reported that the protein content of soybeans is higher at lower latitudes; the accumulation of protein in soybean seeds depends on the interaction between the variety and the cultivation environment, and protein accumulates to higher levels when the plant is matured at a low temperature during the reproductive growth period [20,21]. Weiss et al. [41] reported that the shorter the fruiting period, the higher the protein content is, and the shorter the growing period, the lower the fat content is. Furthermore, other components of soybean seeds vary depending on the variety and cultivation environment. In our study, the soybean varieties originating from the six countries were grown in the same cultivation environment, and the significant difference in protein content of germplasms by country was due to variations in the soybean varieties rather than the cultivation environment.

Park et al. [42] reported that the average total isoflavone content of 106 soybean varieties in Korea was 1489.0 $\mu g \ g^{-1}$, ranging from 527.9 $\mu g \ g^{-1}$ to 3436.5 $\mu g \ g^{-1}$, with the boseokong variety having the highest concentration. In other countries, the variation in isoflavone content among varieties was reported to be 1160–3090 $\mu g \ g^{-1}$ [26,43,44]. However, the average total isoflavone content of the soybean germplasms collected in the six countries was 888.8 $\mu g \ g^{-1}$, in the range of 207.0–3561.8 $\mu g \ g^{-1}$. This result shows that soybean germplasms had 600.2 $\mu g \ g^{-1}$ less than the average total isoflavone content of the Korean varieties. Therefore, the total isoflavone content was applied as an important factor in the soybean breeding programme. In addition, the average total content of isoflavones in 66 varieties of Korean soybean sprout was 1209 $\mu g \ g^{-1}$, in the range of 247–2256 $\mu g \ g^{-1}$ [45]. The average total isoflavone content of Korean bean sprout varieties and the small seed soybean germplasms were similar, at 1156.0 $\mu g \ g^{-1}$ and 267.7–2271.1 $\mu g \ g^{-1}$. According to the results, the total isoflavone content is not an important factor in the development of bean sprout varieties. The variation in isoflavone content with seed colour was the same as seen in a previous study [46]; i.e., yellow seeds (373–1610 $\mu g \ g^{-1}$) contained higher concentrations than black seeds (498–1145 $\mu g \ g^{-1}$) (Table 7).

The concentration of isoflavone in soybeans increases at low temperatures during the maturing stage [36,44], and complex environmental factors, such as average temperature and diurnal temperature difference, are involved in the changes in the isoflavone contents at high altitudes [47]. Paucar-Menacho et al. [48] conducted a comparative analysis between low-protein variety and high-protein varieties of soya. In the analysis, the protein and isoflavone contents showed a strong negative correlation. However, the correlation coefficient between protein and total isoflavone contents in the current study was −0.2200, and there was a weak negative correlation between the contents of protein and each isoflavone component (Figure 3). The correlation between the protein and isoflavone contents is highly negative when only low-protein and high-protein varieties are compared, but soybean germplasms are resources with high amounts of protein and isoflavones. This study has provided useful information on differences in soybean protein and isoflavone contents according to size, colour and country of origin. The findings provide a reference for those selecting genetic resources for the development of high-quality soybean varieties.

## 5. Conclusions

In a comparison of protein content according to country of origin and seed size, there were highly significant differences. In the comparison of isoflavone content according to seed sizes and colour, there were significant differences in the DART results. Daidzein, glycitein and genistein isoflavone contents were positively correlated, while for total isoflavone and protein content, the coefficient was −0.2200. Therefore, total isoflavone and protein showed a low negative correlation.

**Supplementary Materials:** The following are available online at https://www.mdpi.com/article/10.3390/agriculture11080735/s1, Table S1: Calibration equations of isoflavone standards, Figure S1: HPLC chromatogram patterns (UV 254nm) of seed isoflavones, Table S2: Analytical conditions of HPLC for isoflavone, Table S3: Maximum, minimum and mean value of protein contents in soybean germplasms, Table S4: Selected accessions for its high protein contents, Table S5: Maximum, min-

imuim and mean value of isoflavone contents in soybean germplasms, Table S6: Selected accessions for its high isoflavone contents.

**Author Contributions:** Conceptualization, H.-S.K.; Methodology, H.-S.K.; Formal analysis, T.-Y.H. and J.-S.L.; Investigation, T.-Y.H. and J.-S.L.; Resources, H.-S.K.; Data curation, T.-Y.H. and J.-S.L.; Writing—original draft preparation, J.-S.L.; Writing—review and editing, T.-Y.H.; Visualization, T.-Y.H.; Supervision, H.-S.K.; Project administration, T.-Y.H.; Funding acquisition, T.-Y.H. All authors have read and agreed to the published version of the manuscript.

**Funding:** This research received no external funding.

**Institutional Review Board Statement:** Not applicable.

**Informed Consent Statement:** Not applicable.

**Data Availability Statement:** Not applicable.

**Acknowledgments:** This work was supported by the research grant of the Chungbuk National University in 2020.

**Conflicts of Interest:** The authors declare no conflict of interest.

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
