# Peer review of "Variation in Protein and Isoflavone Contents of Collected Domestic and Foreign Soybean (Glycine max (L.) Merrill) Germplasms in Korea"

_agriculture, doi:10.3390/agriculture11080735_

Round 1

Reviewer 1 Report

Introduction should contain more recently references.

Line 44 - 47: These sentences are confused, are contradictory, please reformulated.

Line 89: Uniformized the units according to the journal' instructions. 

Line 119: HPLC analysis conditions must be described on tis section, not in table as supplementary material.

Line 134: Mistake on 99,33%, must be 99.33%. The number of decimal should be the same trough the manuscript.

Protein content are in % and the isoflavone content µg g-1. Please uniformized

Line 140: Replace p-value <0.0001 by p <0.001, and in all manuscript.

Fig. 1, Fig. 2, Table 5, Table 6, Table 7: Include a space before the parentheses

There are two Figure 2.

Footnote of Table 3, replace 100 seed weight: 24g< by 100 seed weight: <24g

Line 279: mistake in "difference". 

Author Response

Line 44 - 47: These sentences are confused, are contradictory, please reformulated.

 Corrected it

Line 89: Uniformized the units according to the journal' instructions. 

Corrected it

Line 119: HPLC analysis conditions must be described on its section, not in table as supplementary material.

 Added HPLC analysis

Line 134: Mistake on 99,33%, must be 99.33%. The number of decimal should be the same trough the manuscript.

Corrected it

Protein content are in % and the isoflavone content µg g-1. Please uniformized

Line 140: Replace p-value <0.0001 by <0.001, and in all manuscript.

Corrected it

Fig. 1, Fig. 2, Table 5, Table 6, Table 7: Include a space before the parentheses

There are two Figure 2.

Corrected it

Footnote of Table 3, replace 100 seed weight: 24g< by 100 seed weight: <24g

Corrected it

Line 279: mistake in "difference". 

Corrected it

Reviewer 2 Report

  1. In row 91 to 92: The harvested soybean germplasms were stored in a -20°C freezer and used for protein and isoflavone analysis. What does germplasms mean here? Seeds, leaves, stems, or whole plants? Need clarify the term germplasm here.
  2. In row 100: analysed by the micro-Kjeldahl method. Here needs a reference
  3. In row 125: All data were subjected to analysis of variance (ANOVA) via the SAS software package. There was NO any report about the ANOVA analysis. The manuscript will be better if a ANOVA table reported in the main text to tell the interactions seed coar color, region, and size.
  4. In row 228 to 231: The correlation coefficients for protein compared to daizein, glycitein, or genistein contents were −0.1694, −0.1706, and −0.222, respectively, and between the protein and total isoflavone content, the coefficient was −0.2200. Therefore, there was a slight negative correlation between the protein content and that of isoflavones. Here is a significant test to show whether the coefficient is significantly different from 0.
  5. The research design is OK. Protein contents and isoflavone concentration vary across years dramatically. the Experiment design can be improved by conducting the experiment for at least 2 years. 

Author Response

1. In row 91 to 92: The harvested soybean germplasms were stored in a -20°C freezer and used for protein and isoflavone analysis. What does germplasms mean here? Seeds, leaves, stems, or whole plants? Need clarify the term germplasm here.

Corrected it

2. In row 100: analysed by the micro-Kjeldahl method. Here needs a reference

Added a reference

3. In row 125: All data were subjected to analysis of variance (ANOVA) via the SAS software package. There was NO any report about the ANOVA analysis. The manuscript will be better if a ANOVA table reported in the main text to tell the interactions seed coar color, region, and size.

Corrected it

4. In row 228 to 231: The correlation coefficients for protein compared to daizein, glycitein, or genistein contents were −0.1694, −0.1706, and −0.222, respectively, and between the protein and total isoflavone content, the coefficient was −0.2200. Therefore, there was a slight negative correlation between the protein content and that of isoflavones. Here is a significant test to show whether the coefficient is significantly different from 0.

5. The research design is OK. Protein contents and isoflavone concentration vary across years dramatically. the Experiment design can be improved by conducting the experiment for at least 2 years. 

Round 2

Reviewer 1 Report

Line 189-190: Mistake on 99,33%, must be 99.33%. The number of decimal should be the same trough the manuscript.

The amounts of protein content should be express as g 100g-1.

Line 140: Replace p-value <0.0001 by p <0.001, and p-value of less than 0.0001 by p< 0.001in all manuscript.

On legends of axis of figures, please, provide space before the units.

Footnote of Table 3 and Table 6 replace 100 seed weight: 24g< by 100 seed weight: <24g

Line 279: mistake in "difference". 

Author Response

Line 189-190: Mistake on 99,33%, must be 99.33%. The number of decimal should be the same trough the manuscript.

Corrected it

The amounts of protein content should be express as g 100g-1.

Corrected it

Line 140: Replace p-value <0.0001 by <0.001, and p-value of less than 0.0001 by p< 0.001in all manuscript.

Corrected it

On legends of axis of figures, please, provide space before the units.

Corrected it

Footnote of Table 3 and Table 6 replace 100 seed weight: 24g< by 100 seed weight: <24g

Corrected it

Line 279: mistake in "difference". 

Corrected it